# Pharmacokinetic–Pharmacometabolomic Approach in Early-Phase Clinical Trials: A Way Forward for Targeted Therapy in Type 2 Diabetes

**DOI:** 10.3390/pharmaceutics14061268

**Published:** 2022-06-15

**Authors:** Khim Boon Tee, Luqman Ibrahim, Najihah Mohd Hashim, Mohd Zuwairi Saiman, Zaril Harza Zakaria, Hasniza Zaman Huri

**Affiliations:** 1Department of Clinical Pharmacy and Pharmacy Practice, Faculty of Pharmacy, Universiti Malaya, Kuala Lumpur 50603, Malaysia; teekhimboon12@gmail.com; 2National Pharmaceutical Regulatory Agency, Ministry of Health Malaysia, Petaling Jaya 46200, Malaysia; zaril@npra.gov.my; 3Department of Medicine, Faculty of Medicine, Universiti Malaya, Kuala Lumpur 50603, Malaysia; luqman@um.edu.my; 4Department of Pharmaceutical Chemistry, Faculty of Pharmacy, Universiti Malaya, Kuala Lumpur 50603, Malaysia; najihahmh@um.edu.my; 5Centre for Natural Products Research and Drug Discovery (CENAR), Universiti Malaya, Kuala Lumpur 50603, Malaysia; zuwairi@um.edu.my; 6Institute of Biological Science, Faculty of Science, Universiti Malaya, Kuala Lumpur 50603, Malaysia; 7Centre for Research in Biotechnology for Agriculture (CEBAR), Universiti Malaya, Kuala Lumpur 50603, Malaysia; 8Clinical Investigation Centre, Universiti Malaya Medical Centre, Kuala Lumpur 50603, Malaysia

**Keywords:** targeted therapies, pharmacokinetics, pharmacometabolomics, pharmacodynamics, early phase clinical trials, metformin, diabetes, metabotypes, precision medicine

## Abstract

Pharmacometabolomics in early phase clinical trials demonstrate the metabolic profiles of a subject responding to a drug treatment in a controlled environment, whereas pharmacokinetics measure the drug plasma concentration in human circulation. Application of the personalized peak plasma concentration from pharmacokinetics in pharmacometabolomic studies provides insights into drugs’ pharmacological effects through dysregulation of metabolic pathways or pharmacodynamic biomarkers. This proof-of-concept study integrates personalized pharmacokinetic and pharmacometabolomic approaches to determine the predictive pharmacodynamic response of human metabolic pathways for type 2 diabetes. In this study, we use metformin as a model drug. Metformin is a first-line glucose-lowering agent; however, the variation of metabolites that potentially affect the efficacy and safety profile remains inconclusive. Seventeen healthy subjects were given a single dose of 1000 mg of metformin under fasting conditions. Fifteen sampling time-points were collected and analyzed using the validated bioanalytical LCMS method for metformin quantification in plasma. The individualized peak-concentration plasma samples determined from the pharmacokinetic parameters calculated using Matlab Simbiology were further analyzed with pre-dose plasma samples using an untargeted metabolomic approach. Pharmacometabolomic data processing and statistical analysis were performed using MetaboAnalyst with a functional meta-analysis peaks-to-pathway approach to identify dysregulated human metabolic pathways. The validated metformin calibration ranged from 80.4 to 2010 ng/mL for accuracy, precision, stability and others. The median and IQR for Cmax was 1248 (849–1391) ng/mL; AUC_0-infinity_ was 9510 (7314–10,411) ng·h/mL, and Tmax was 2.5 (2.5–3.0) h. The individualized Cmax pharmacokinetics guided the untargeted pharmacometabolomics of metformin, suggesting a series of provisional predictive human metabolic pathways, which include arginine and proline metabolism, branched-chain amino acid (BCAA) metabolism, glutathione metabolism and others that are associated with metformin’s pharmacological effects of increasing insulin sensitivity and lipid metabolism. Integration of pharmacokinetic and pharmacometabolomic approaches in early-phase clinical trials may pave a pathway for developing targeted therapy. This could further reduce variability in a controlled trial environment and aid in identifying surrogates for drug response pathways, increasing the prediction of responders for dose selection in phase II clinical trials.

## 1. Introduction

Metabolites are the molecules that react in metabolic reactions in a living organism and constantly change in a myriad of chemical reactions [1]. Metabolomics is the study of endogenous and exogenous metabolites in a biological system using emerging technologies, such as liquid chromatography or gas chromatography–mass spectrometry and NMR [2]. The application of metabolomics through measuring the metabolic profiles of drug reactions and drug variation responses in the biological system is defined as pharmacometabolomics [3,4]. The drug pharmacology effects interplay with the epigenetic factors, environmental factors, demographic characteristics and disease-related factors [5]. Identifying the dysregulated human metabolic pathways in pharmacometabolomic study helps to clarify the multifaceted pharmacological effects: mechanism of actions, safety biomarkers and efficacy biomarkers.

Pharmacokinetic studies measured the rate and extent of a drug’s absorption, distribution and elimination in the body [6], which provides information on a drug’s Cmax (maximum plasma concentration) and Tmax (time to reach Cmax) and other parameters. The integration of pharmacometabolomic study with pharmacokinetics and pharmacodynamics specifically studies the temporal changes in drug concentration, and endogenous metabotypes were proposed to realize personalized medicine [5]. It is postulated that the drug was bound to most of the target site at the peak plasma concentration to trigger the highest pharmacodynamic changes in the therapeutic dose; therefore, the pharmacometabolomics analyzed the individual metabolic profiles between the baseline and the treatment data’s potential, revealing the drug’s multifaceted pharmacological effects in therapeutics and adverse drug reactions.

Metformin has been a first-line antidiabetic agent for decades, but the mechanism of action remains unclear. The pharmacological effects of metformin include increasing insulin sensitivity and glucose uptake into cells, inhibiting hepatic gluconeogenesis and improving glucose update and utilization [7]. Besides its antidiabetic effects, metformin is also used for weight reduction, lowering plasma lipid levels, prevention of vascular complications and treatment of polycystic ovary syndrome [8]. Metformin treatment is linked to the tricarboxylic acid (TCA) cycle, urea cycle, glucose metabolism, lipid metabolism or gut metabolism. Pharmacometabolomic research on metformin is scarce, yet identification of the metabolic changes that affect variation of the pharmacodynamics of metformin is critical to achieving the desired therapeutic outcomes [9].

Many in vitro and in vivo non-clinical studies have investigated the pharmacological effects of metformin associated with complex I inhibition, which leads to 5′ AMP-activated protein kinase (AMPK) activation using a supra-pharmacological metformin concentration [10]. Several metformin pharmacometabolomic studies were performed on healthy volunteers and patients to investigate type 2 diabetes [11], obesity [12], antitumor [13], metabolic syndrome [14] and cardiovascular risk [15]; one study identified the metabolic changes of metformin based on the pharmacokinetics for three time-points around Cmax and 36 h in serum samples [11].

In the early-phase of clinical drug development, pharmacometabolomics could contribute to identification of the mechanism of drug response variations, elucidate safety and efficacy biomarkers, aid in patient selection and contribute to late-phase trial design. [16]. The United States Food and Drug Administration identified potential biomarkers that can be submitted in the new drug applications process. The biomarker is categorized as diagnostic, monitoring, predictive, prognostic, pharmacodynamic or response, safety and susceptibility [17]. Pharmacometabolomics is a useful tool to investigate the baseline and treatment metabotypes in early-phase clinical trials to identify the potential biomarkers [4]. In this proof-of-concept study, metformin was selected as a model drug to explore the pharmacokinetics and pharmacometabolomics in early-phase clinical trial settings to identify the perturbation of human metabolic pathways using a single dose of metformin in healthy subjects.

## 2. Materials and Methods

### 2.1. Study Design

The study is a prospective, open-label, single-dose oral administration of metformin 1000-mg tablets in healthy subjects under fasting conditions (at least 10 h before dosing) conducted at the Clinical Investigation Centre, University Malaya Medical Centre. The study was approved by the Ethics Committee (MEC ID 2018112–6848) and registered with clinicaltrials.gov (Identifier ID: NCT04161404).

Subjects were screened for a list of inclusion and exclusion criteria: non-smoking males between 18 and 45 years old with body mass index of 18.5–20.5 kg·m^2^. Subjects were excluded for a list of illnesses and clinically significant abnormal laboratory testing. Subjects were restrained from taking over-the-counter medication 14 days before the dosing and herbal remedies or caffeine drinks 7 days before the dosing. Seventeen eligible volunteers were administered a single dose of metformin 1000 mg after 8 h of fasting. Fifteen blood samples (0, 0.5, 1, 1.5, 2, 2.5, 3, 3.5, 4, 5, 6, 8, 10, 12, 24 h) and four urine samples (U0: pre-dose, U1: 0–4 h, U2: 4–8 h and U3: 8–12 h) were collected from the subjects.

Three cohorts were planned to ensure sufficient clinical ward staff and space to monitor every subject on the dosing day. Subjects were housed in a controlled environment, and standardized meals were provided to each cohort. The blood samples were collected using an ethylenediaminetetraacetic acid (EDTA) tube and centrifuged at 10,000 rpm for 10 min at 4. The plasma samples together with the urine samples were kept in a −80 °C freezer in cryovials for further analysis. The subjects were monitored for adverse events until seven days post-dose. Pre-dose and peak-concentration plasma samples were used for untargeted pharmacometabolomic analysis to determine the treatment metabotypes.

### 2.2. Bioanalytical

The reference-standard metformin (Batch 3267, purity 99.8%) was sourced from British Pharmacopoeia, London, UK. Methanol, acetonitrile, acetone, acetic acid and LCMS-grade water were purchased from Merk (Darmstadt, Germany). The column and guard column were obtained from Agilent (Santa Clara, CA, USA): stainless steel guard column, Zorbax-SB-C8 Rapid resolution cartridge (2.1 × 30 mm 3.5 µm) (873700-936), rapid resolution cartridge holder and hardware kit (820555-901) and separation column Zorbax SB-Aq 1.8 µm 2.1 × 50 mm (827700-914). Nylon filters with a size of 0.22 µm were used for all the samples in LCMS sample preparation.

All samples were analyzed with an ultrahigh performance liquid chromatography system coupled with a high mass accuracy tandem quadrupole time-of-flight mass spectrometry (UPLC-QTOF-MS) (Agilent Technologies, Santa Clara, CA, USA) based on modified METLIN methods [18] in electrospray ionization (ESI) positive mode and negative mode, respectively. Chromatographic separation was performed using a Zorbax-SB-C8 guard column (2.1 × 30 mm 3.5 µm) and separation column Zorbax SB-Aq 1.8 µm 2.1 × 50 mm (Agilent Technologies, Santa Clara, CA, USA). The mobile phase was solvent A (water with 0.2% *v*/*v* acetic acid) and solvent B (methanol with 0.2% *v*/*v* acetic acid) with a gradient system: 0–13 min, 98% to 2% A; 13–19 min, 2% A with 5-min post-run. The flow rate was 0.6 mL/min, and the injection volume was 10 µL for untargeted metabolomic analysis in positive and negative modes. For pharmacokinetic analysis, a similar flow rate, injection volume and mobile phase were applied, and the gradient system was shortened to 0–0.5 min, 99% A; 0.5–3, 99% to 1% A; 3–5 min, 1% A with 2-min post-run.

The mass spectrometry parameters were set as gas temperature 290 °C, gas flow 11 L/min, nebulizer 45 psig, sheath gas temperature 350, sheath gas flow 11 L/min, fragmentor 140, skimmer 65 and octupole RF Peak 750; the mass range was set at 50–1100 *m/z*. In positive ion mode, the VCap was set at 4000, and the Nozzle voltage was at 0 V. In negative ion mode, the Vcap was 3500, and the Nozzle voltage was at 1000 V. Reference masses used in the QTOF were 121.05087300 and 922.00979800 for positive ion mode and 68.99575800, 112.98558700 and 1033.98810900 for negative ion mode.

For pharmacokinetic analysis, the quantitative LCMS method was validated using metformin spiked with the blank plasma. A series of metformin concentrations were spiked into 200 µL of blank plasma. The plasma was spiked with 600 µL of acetonitrile, acetone and methanol (1:1:1) and incubated at −20 °C freezer for one hour. The samples were centrifuged at 10,000 rpm at 4 °C for 15 min and filtered with 0.2 µm nylon filter into LCMS vials.

In the untargeted metabolomic sample processing, the plasma sample preparation is similar to the above pharmacokinetic plasma sample preparation. For urine sample preparation, 200 µL subject samples were added with 600 µL acetonitrile, acetone and methanol (1:1:1) and centrifuged at 10,000 rpm at 4 °C for 15 min. The aliquot was mixed with LCMS grade water (1:1 ratio) and filtered.

The plasma or urine sample processing batch was run for positive mode and negative mode separately. The samples’ sequence arrangements started with a blank, blank with internal standards, blank plasma and blank plasma with internal standards; six pool quality control (PQC) samples were followed by the subject samples. The subject samples were interspersed with PQC samples for every four subject samples until the end of the batch run (Appendix A). The time-points’ samples were interspersed, but the subject’s numbers were according to sequence throughout the analysis run. The sequence arrangement for this untargeted metabolomic analysis was arranged according to the subject’s number followed by time-points. The untargeted metabolomics could be improved with randomization of the sequence to avoid instrumental or technical bias [19,20].

### 2.3. Method Validation and Statistical Analysis for Pharmacokinetics

The metformin quantitative analytical methods were validated based on bioanalytical method validation guidelines [21] for between run and within run accuracy and precision, selectivity, recovery, carryover and stability in the short term, autosampler, three freeze-thaw cycle and long-term stability. Metformin was found in the positive mode only at 130.1086 *m/z* (Appendix A). Seventeen subjects’ plasma samples (15 timepoints each subject) were analyzed using the validated method to obtain the metformin concentration at each time-point. In the pharmacokinetic analysis, metformin pharmacokinetic parameters for maximum plasma concentration (Cmax), area under the plasma concentration time-curve (AUC), time to reach Cmax (Tmax), half-life and clearance were calculated using Matlab SimBiology software with the non-compartmental model (Figure 1).

### 2.4. Data Processing and Statistical Method for Untargeted Metabolomics

The plasma metabolomics analysis was carefully designed by dividing the samples into two batches for pharmacometabolomic exploration. The first batch (dataset A) investigated the significant different metabolites for pre-dose (T0) versus times 2.5, 3 and 3.5 h for six subjects to observe the number of significant compounds in LCMS positive mode and negative mode (Figure 1). The median Tmax for metformin was established at 2.5 h, the three plasma samples were selected with the assumption that the pharmacological effects of metformin reached maximum dose effects within 2.5 to 3.5 h. This strategy could provide alternative methods to reduce the sample size (*n* = 6, 18 paired analyzed samples) and increase the coverage of metabolites in peak plasma duration. The second batch (dataset B) utilized pharmacokinetic-guided maximum plasma time- point samples and pre-dose to identify the significant metabolites. This method has the advantage of a high biological sample size (*n* = 17, 17 paired analyzed samples) and an individualized peak plasma concentration for the determination of human metabolic pathways. For the metabolomic urine samples, six subjects for four time-points (U0, U1, U2 and U3) were analyzed in positive mode and negative mode.

The general statistical analysis and data processing for the pharmacometabolomic was performed using MetaboAnalyst (Figure 1). First, the chromatograms were converted into mzXML file using Global Natural Product Social Molecular Networking (GNPS) software. Zipped files were further processed using the MetaboAnalyst 5.0 software, which provides end-to-end services from spectral processing to pathway prediction [22,23]. The raw spectral datasets were then processed in the MS Spectra Processing module, followed by normalization in the statistical module, continued with batch correction module and back to the statistical module for multivariate analysis in positive mode and negative mode. Unsupervised principal component analysis (PCA) was first applied to observe the features’ separations among the time-points. Additional supervised partial least square discriminant analysis was applied when the pattern of separation in PCA is not clear; cross validation and permutation tests were performed to test for overfitting. There are two common metabolomic analysis methods in the following steps: Individual peak annotation focuses on a single compound or functional pathway prediction based on the mummichog algorithm’s focus on individual pathways [22]. Functional pathway analysis bypassing the identification of the compounds was performed in this study to identify the dysregulated biological pathway.

In this untargeted metabolomic analysis, gliclazide and atenolol were used as the internal standard in the LCMS analysis. Gliclazide was detected in both the positive and negative modes, but atenolol was only found in positive mode. Gliclazide’s signal consistency in six analysis batches was visually checked after MS Spectral Processing. Appendix A shows representative boxplots (positive and negative mode) for abundance in logarithm base 2 (log2) intensity of gliclazide. The log2 intensity in positive mode at 324.1383 *m/z* is between 20 and 22 and the negative mode at 322.1239 *m/z* is between 17 and 20. The abundance has a slight variation in log2 intensity. In the data normalization, gliclazide was selected for probabilistic quotient normalization, data transformation using logarithm 10 and data scaling using range scaling and Pareto scaling. Sample normalization (row-wise) using the internal standard aims to remove systematic variation between experimental conditions unrelated to the biological differences and feature normalization (column-wise), which includes log transformation and range scaling, bringing variances of all features close to equal.

The PQC samples were initially visually inspected to ensure consistency before they were subjected to sample analysis. After the analyses were completed, the quality control features were checked for every chromatogram. The batch correction module in MetaboAnalyst 5.0 [24] provides several algorithms based on feature characteristics and data types. Eigen MS batch correction was applied in dataset A and COMBAT batch correction was applied in dataset B. Appendix A shows the PCA diagrams of the features before adjustment and after adjustment and the bar chart comparison of the distance of features between the original data and adjusted data. The features were dispersed evenly in the matrix after adjustment in the PCA based on the algorithm with the shortest distance among the original, Eigen MS [25] and Combat [26].

Further multivariate analysis was performed using the PCA for different time-points’ visualizations in each treatment group. After this, a functional meta-analysis module was applied to time-points’ difference effects in the three treatments for putative identification of metabolites based on the human Kyoto Encyclopedia of Gene and Genome (KEGG) library. This functional meta-analysis pooled all MS peaks, bringing out the weaker signals in selected datasets. Data heterogeneity was adjusted based on the MS ionization mode and accuracy of the LC-MS instrument during the putative metabolite annotation in the program. The phenotype effects from humans were identified through these robust meta-signatures from multiple datasets at different time-points. 

First, this module started with a dataset upload from MS spectral processing, normalization with log transform, median and auto-scaling box plots and data analysis using a t-test. The mass tolerance was set at 15 ppm for all datasets, and the p-value cutoff point was adjusted between 0.001 and 0.005 based on the recommendation from the mummichog algorithm to achieve 10–25% of significant features [27]. Second, the pooling peaks method was chosen to improve the metabolome coverage by combining complementary measurements from the available datasets. Another reason for using pooling peaks was that the same samples were analyzed in positive mode and negative mode. The positive and negative peaks were merged into a single dataset for compound annotation and predicted the pathway activities. In this step, the mummichog algorithm was chosen, and version 1 was applied (consideration of *m/z* features based on adducts, *p*-value and ionization mode) with *p*-values cutoff at 0.001 using *Homo sapiens* KEGG library. Last, a list of perturbed human metabolic pathways was generated from MetaboAnalyst. Appendix A shows the normalization parameters and the number of features in the data processing batch.

## 3. Results

### 3.1. Clinical Trial Results

A total of 30 subjects were screened for eligibility; ten subjects who did not meet the inclusion criteria or met the exclusion criteria were excluded, and two subjects withdrew their consent. Eighteen subjects were scheduled for metformin 1000 mg in three cohorts; one subject did not attend the dosing day in the third cohort and withdrew from the study. Seventeen subjects completed the follow-up, and the samples were analyzed (Figure 2). Patient demographic data and clinical characteristics are presented in Table 1. One subject experienced abdominal pain after the dosing. From the clinical laboratory characteristic, the data were within normal range and considered no significant changes based on clinician judgement. The glucose monitoring for pre-dose and the first four hours after dosing were within normal range and demonstrated no hypoglycemic effects in healthy volunteers (Appendix A). One adverse drug reaction occurred in one subject three hours after the dosing, which is considered gastrointestinal intolerance by the investigator. A total of 255 plasma samples were injected into LCMS for pharmacokinetic analysis; 58 plasma samples and 24 urine samples were run in LCMS for pharmacometabolomic analysis.

### 3.2. Metformin Analytical Method Validation

The LCMS bioanalytical method was validated based on several parameters according to the bioanalytical method validation guidance [21]. A calibration curve was established between 80.4 and 2010 ng/mL with low, medium and high-quality control samples at 100.5, 140.7 and 703.5 ng/mL (Appendix A). The results of the method validation parameters for accuracy, precision, carry over, recovery, selectivity and stability in bench top, three freeze-thaw cycles, auto-sampler and long-term stability are presented in Table 2. The method validation detail data for the parameters are shown in Appendix A. 

### 3.3. Pharmacokinetics Profiles

The above-validated method was applied to analyze subjects’ plasma samples. A total of 255 plasma samples (*n* = 17) were analyzed and quantified for metformin concentration (Appendix A). Pharmacokinetic analyses were performed using non-compartmental analysis in the Matlab R2021b – SimBiology version 6.2, The MathWorks, (California, US). Individual pharmacokinetic parameters are presented in Appendix A. Table 3 demonstrates the pharmacokinetic parameters for oral administration of metformin 1000 mg in healthy subjects. The median for Cmax was established at 1248.1 ng/mL; Tmax was 2.5 h, and the half-life was 6.8 h for the healthy subjects.

The times to reach the peak plasma concentration were 2.5, 1.5, 2.5, 3, 2, 1.5, 2, 2.5, 3, 2.5, 2.5, 3, 2.5, 4, 2, 4 and 2.5 h, respectively (Figure 3). These individualized peak plasma concentration samples guided the selection of samples for the pharmacometabolomic analysis.

### 3.4. Metabolomics Analysis of Metformin in Plasma and Urine Samples

#### 3.4.1. Metabolomic Multivariate Analysis

The application of a multivariate analysis aims to reduce dimensionality. The PCA for the first batch dataset A in positive mode and negative mode is shown in Figure 4. The time-points for T0 and the groups of T2.5, 3 and 3.5 h have clear separation patterns in positive mode PCA but not in negative mode PCA. Both PCAs show clear overlapping for peak time-points T2.5, T3 and T3.5 (dark blue, light blue and pink) and separation with time-point T0 (green); this could be different phenotypes of the metformin metabolism in pre-dose and peak dose. The quality control samples (red) were scattered around the center of the PCA. The negative mode data were further analyzed using a supervised partial least square discriminant analysis (PLS-DA), which demonstrated significant separation for the above two groups. Cross-validation (Q2 = 0.540, R2 = 0.948) and a permutation test (*p* < 0.05) were used to evaluate overfitting of the model.

Similar multivariate analyses for urine samples in four time-points were visually inspected. The PCA was based on the duration of urine collection in the positive and negative modes to explore significant compounds and significant pathways. Figure 5 demonstrated that most of the quality control (QC) samples were primarily scattered in the center. At the same time, the other time-points were separated into different regions in this unsupervised method.

#### 3.4.2. Metabolomic Functional Pathway Analyses

The human metabolic pathways with the number of metabolites in the Kyoto Encyclopedia of Gene and Genome (KEGG), total metabolite hits, significant metabolite hits based on mummichog algorithm for pre-dose versus T2.5, T3, T3.5 h (dataset A) and pre-dose versus peak-dose samples (dataset B) as well as pre-dose versus 0–4 h post-dose urine samples were presented in Table 4. A total of 14 provisional dysregulated human metabolomic pathways were observed from the plasma samples in dataset B (*n* = 17, 17 pairs pre-dose versus peak-dose data); a total of 11 out of 14 provisional dysregulated human metabolic pathways were also found in the plasma sample dataset A (*n* = 6, 18 pairs pre-dose versus T2.5, 3, 3.5 h data) and urine sample dataset U1 (*n* = 6, pre-dose versus 0–4 h data). An increasing number of metabolites hit the KEGG pathways when a higher number of subjects were analyzed in dataset B compared to dataset A. However, analyses using a lower number of subjects with three samples around the peak plasma concentration provides broader insights for the number of human metabolic pathways. Three provisional dysregulated human metabolic pathways (riboflavin metabolism, retinol metabolism, glycerophospholipid metabolism) were found significantly in dataset A and dataset B.

There were 37 metabolites for arginine and proline metabolisms in the KEGG library; the mummichog analysis found an increasing trend in the number of metabolites to hit the pathway from dataset A (24 metabolites) to dataset B (31 metabolites) when the number of subjects increases from 6 subjects to 17 subjects. The urine dataset also found 28 metabolites hit the pathway.

Table 5 demonstrated the predicted metabolic pathways found in the three time-point groups in metformin 1000-mg dosing. Arginine and proline metabolism, butanoate metabolism and arginine biosynthesis were found in time-point U0 versus U1 and U0 versus U2 groups. The total metabolite hits were slowly decreased over time for the above metabolic pathway, suggesting that the effects of metformin peaks during the first four hours (U0 versus U1), and it slowly decreases from four to eight hours (U0 versus U2). The significant metabolite hits for the arginine and proline metabolisms’ metabolic pathways were creatine, gamma-aminobutyric acid, 4-Aminobutyraldehyde, L-4-Hydroxyglutamate semialdehyde and L-Glutamic acid.

From the pharmacometabolomic analysis, the metformin metabolites obtained from the boxplot of metformin plasma samples (dataset A) and metformin urine samples in positive mode during MS spectral processing provided additional information about the kinetics of metformin in healthy subjects (Appendix A). In the metformin plasma dataset A, metformin intensity was not observed at 0 h; the concentration increased at time-points 2.5, 3 and 3.5 h, a reducing trend happened at 8 h and a return to zero at 12 h. The information was consistent with the pharmacokinetic data. In the urine samples, metformin was not found at 0 h, achieved high concentration in the first 8 h and a slight reduction during 8–12 h.

## 4. Discussion

### 4.1. Clinical Trial

This is a proof-of-concept study to explore the pharmacodynamic effect of metformin through metabolomics based on the metformin maximum plasma time concentration of an individual subject. The phase one pharmacological study in an early-phase clinical trial normally employs six to nine subjects for each dose [28]. The 3 + 3 phase one trial design is the gold standard [29], but 6 + 14 for dose expansion was recommended in the cancer trial [30]. For metabolomic analysis without experimental pilot data, 12 subjects were proposed based on a dynamic probabilistic principal component analysis or 18 subjects in each group for probabilistic principal components and covariates analysis [31]. Metabolic profiles obtained before, during and after drug administration could provide insights into the mechanism of action and variation response to the drug treatment [4]. Therefore, a single-arm strategy was applied in the study design to focus on the baseline and treatment metabotypes to discover both inter-patient and intra-patient variations in drug response. This study investigates the provisional dysregulated human metabolic pathways using metabolomic analysis; eighteen subjects were planned, and seventeen subjects completed the trial.

One subject (S12) recorded stomach cramps three hours after the dosing; the adverse drug reaction occurred after metformin reached Tmax (3 h for subject 12) and a higher range of Cmax (1423.4 ng/mL) based on a personalized pharmacokinetic profile. The adverse drug reaction is consistent with common gastrointestinal intolerance side effects of the metformin tablet [32].

### 4.2. Pharmacokinetics

The Cmax (1248 ng/mL or 9.662 µM) and AUC_0-infinity_ (9510 ng·h/mL), Tmax (2.5 h) and half-life (6.8 h) in this study are consistent with the pharmacokinetics and a bioequivalence study using the same formulation [33]. The results demonstrated that the pharmacokinetic profiles for metformin in Caucasians are similar to the Asian population. These data may combine with other pharmacokinetic studies to determine bioequivalence through a network meta-analysis [34], which could provide additional information about the interchangeability of brand and generic products. The plasma concentration of metformin reached 25 µM within three hours of oral administration of 1000 mg of metformin in non-diabetic subjects. Diabetic patients who were administered 1000 mg of metformin twice daily chronically achieved peak plasma concentrations of approximately 40 µM. The therapeutic range of plasma metformin concentration in humans is between 10 and 40 µM. Most of the mitochondrial complex I inhibition leading to AMPK activation were using the supra-pharmacological metformin concentration (>1000 µM), which does not occur in clinical setting [10]. Here, the clinical therapeutic dose was focused on to explore the pharmacometabolomic effects based on metformin plasma concentration.

### 4.3. Metabolomics in Plasma and Urine Samples

From the results, both untargeted metabolomic strategies using 6 subjects (dataset A) and 17 subjects (dataset B) provided similarly significant dysregulated biological pathways; the second strategy comprised slightly increased total metabolite hits than the first strategy. However, the first strategy could be the more useful pharmacometabolomic method for the current pharmacokinetics profiled in phase one single-dose escalation or multiple-dose escalation studies, which commonly recruit six to nine subjects for a single dose. The urine dataset (6 subjects) consists of three paired sub-datasets (U0 versus U1, U0 versus U2 and U0 versus U3) for exploration of the pharmacodynamic-related metabolic pathways’ profiling based on the pharmacokinetics of metformin.

#### 4.3.1. Arginine and Proline Metabolism

The highest significant compounds observed from all the datasets were arginine and proline metabolism; significant compounds observed were L-proline, D-proline, S-adenosylmethioninamine, N-acetylputrescine, creatine, gamma-aminobutyric acid, 4-aminobutyraldehyde, L-4-hydroxyglutamate semialdehyde and L-glutamic acid (Table 4). In the global urine metabolomic analysis according to different time-points, the metabolite hits for the arginine and proline metabolism was highest in U1 (28 metabolites), slowly decreased in U2 (26 metabolites) and not identified in U3 (Table 5). A similar metabolomic study conducted on non-diabetic subjects demonstrated that the arginine and proline metabolism was a significant pathway found using untargeted metabolomics for plasma samples at 12.30 h after the first dose of metformin and 2 h after the second dose of metformin [35]; the significant compounds identified were L-Aspartic acid, citrulline, L-glutamic acid and ornithine. The down-regulation of arginine in type 2 diabetes patients taking metformin was found. [12]. A negative correlation between some aliphatic amino acids was associated with insulin sensitivity and type 2 diabetes [36]. Dysregulation of the arginine and proline metabolism with metformin intervention might be associated with insulin sensitivity.

#### 4.3.2. Valine, Leucine and Isoleucine Biosynthesis

The branched-chain amino acid (BCAA) was also observed from the valine, leucine and isoleucine biosynthesis in dataset A and dataset U1 with significant compounds 3-Methyl-2-oxovaleric acid, L-Leucine and L-Isoleucine. The results are similar to a metabolomics study that used single-dose metformin 500 mg, whereby valine, leucine and isoleucine biosynthesis were the most significant changes in the biochemical pathways [11]. BCAA is the potential biomarker of diseases, such as insulin resistance and type 2 diabetes; it functions as a regulator of energy homeostasis, glucose and lipid metabolism, gut health and immunity [37]. The aminolyacyl-tRNA biosynthesis with significant compounds of L-Proline, L-Tryptophan, L-Isoleucine, L-Leucine and L-Glutamic acid was found in three datasets. A non-clinical study using healthy mice focused on metformin’s effects on altered gut microbiota also found that aminolyacyl-tRNA biosynthesis is significantly enriched [38]. In the Copenhagen Insulin and Metformin Therapy trial study, the effect of metformin’s plasma metabolite profile found elevated leucine or isoleucine levels demonstrating the possible metabolic changes after administration of metformin [39].

#### 4.3.3. Glutathione Metabolism

Type 2 diabetes patients were demonstrated to have lower glutathione [40], specifically with microvascular complication [41]. Oral metformin treatment changed the glutathione level in diabetic rats [42]. A nonclinical study in rats showed metformin ameliorated inflammation of the pancreas through modulation of the JAK/STAT pathway to restore glutathione status and inhibit proinflammatory cytokines [43]. The perturbed glutathione metabolism in the plasma and urine datasets could be linked to the JAK/STAT signaling pathway in anti-diabetic effects.

#### 4.3.4. Galactose Metabolism

The results showed that mannotriose and raffinose were the significant compounds from the plasma samples, and D-Galactose, alpha-D-Glucose, D-Galactose, D-Glucose, D Fructose, D-Mannose and Myo-inositol were the significant compounds from the urine samples in metformin dysregulation of galactose metabolism. Raffinose was found to increase Glut4 translocation via phosphorylation of IRβ/PI3K/Akt in differentiated L6 myocytes and 3T3-L1 preadipocytes. It is potentially involved in glycogen synthesis by inhibiting the activation of GSK3β, which is associated with increased insulin sensitivity [44]. Raffinose also demonstrated induced lipid oxidation with a simultaneous reduction in the lipid synthesis [45]. Increased sensitivity of insulin and lipid metabolism effects could be the effects of dysregulation of raffinose in the galactose metabolism.

#### 4.3.5. Tryptophan Metabolism

Dysregulation of tryptophan and kynurenine is associated with the mechanism of insulin-resistance [46]. The three datasets above demonstrated that more than two-thirds of the metabolites hit the tryptophan metabolism pathway in KEGG; the identified compounds are L-Tryptophan, 5-Hydroxy-N-formylkynurenine and 5-Hydroxy-L-tryptophan. A study has demonstrated that kynurenic acid was increased in the plasma of type 2 diabetes patients [47]. Metformin was found to restore insulin sensitivity by down-regulation of the kynurenine pathway metabolism [48]. The mechanism of action caused by the kynurenine metabolites was the formation of chelate complexes with insulin that has a 50% reduction in activity compared to the insulin [49]. Normalization of tryptophan metabolism by metformin reduced the kynurenine metabolic pathway, which is associated with the reduction in insulin resistance.

#### 4.3.6. Retinol Metabolism

Retinol metabolism was observed in both plasma metabolomics datasets, with significant compounds of β-Carotene and retinoyl b-glucuronide. Retinol and retinol-binding protein 4 (RBP4) were associated with type 1 diabetes [50,51] and type 2 diabetes [52]. RBP4 is responsible for transporting retinol from the liver to the peripheral [53]. Elevation of RBP4 is linked to multiple insulin-resistant mice models; RBP4 reacted by inducing the expression of gluconeogenic enzyme phosphoenolpyruvate carboxykinase in the liver to impair insulin signaling in the muscle [54]. A study conducted on type 2 diabetes patients also demonstrated that serum RBP4 levels were associated with insulin resistance and severity of coronary artery disease [55]. Metformin intervention in an in vivo study demonstrated a decrease in RBP4, thereby improving the insulin sensitivity [56].

#### 4.3.7. Starch and Sucrose Metabolism

From both of the analysis strategies in dataset A and dataset B, the starch and sucrose metabolism were observed with 13 metabolite hits, and dextrin was the significant compound. It was found that the number of metabolites from both strategies hit all the metabolites from the starch and sucrose metabolism in the KEGG pathway. The metformin tablet contains excipients, such as sodium starch glycollate, maize starch, povidone, colloidal anhydrous silica and magnesium stearate; the starch and sucrose metabolism could be the effects of consuming the excipients in the metformin tablet [57]. This explains the holistic effects of the emerging metabolomics technologies, which can not only identify the phenotypic effects of metformin’s active ingredient but also the effects of metformin’s tablet dosage form. Additional study is required to differentiate the action between metformin and the excipient in the tablet formulation.

#### 4.3.8. Glycosaminoglycan Degradation

Glycosaminoglycan degradation was observed in the U0 versus U1 group only. (GalNAc)2 (GlcA)1 (S)1, (GlcA)1 (GlcNAc)1 (S)1 and Chondroitin 4-sulfate were significant metabolites hit in the glycosaminoglycan degradation metabolic pathways. The perturbed glycosaminoglycan degradation metabolic pathways correspond to a study using a sulphated glycosaminoglycan assay kit in diabetes patients treated with metformin for six months of urinary samples [58]. This effect was observed in the first four hours of urine samples for single-dose oral administration of metformin in healthy volunteers and first-morning urine samples for six months of monotherapy of metformin in type 2 diabetes patients. The clinical outcome was not observed in the single-dose administration of metformin. However, the perturbed biological changes were observed in the single-dose and six-month treatment duration for the metformin therapy. Degradation of glycosaminoglycan reduces the non-enzymatic degradation of the glycans, which contributes to prevention of vascular diabetes-related complications.

### 4.4. Application of Pharmacometabolomics in Clinical Drug Development

In recent years, applications of pharmacometabolomics were broadly reviewed, specifically in data processing and statistical analysis [59], biomarker discoveries and precision medicine [4,60], early-phase clinical development [16] and pharmacology studies [3,61]. A model was proposed to describe the potential framework of pharmacometabolomics in clinical drug development (Figure 6).

In the early drug development process, the context of use for biomarkers can be submitted during investigational new drug applications. Pharmacometabolomics could be initiated in a phase I clinical trial through the integration of a personalized pharmacokinetic and pharmacometabolomic approach, which has the advantage of reducing the study population’s variability in a controlled environment and aiding identification of surrogate drug response pathways from the provisional dysregulated human metabolic pathways. As the clinical drug’s development continues with single ascending dose and multiple ascending dose studies, provisional dysregulated human metabolic pathways may lead to the identification of a drug’s mechanism of action, while the provisional predictive biomarkers help to reveal various drug response biomarkers.

Identification of the provisional diagnosis or treatment response associated with biomarkers increases the prediction of responders for dose selection in phase II trials which can lower the cost and shorten the drug development process. The provisional therapeutic biomarkers associated with the clinical efficacy and safety of a drug can be validated at this stage.

After marketing authorization, besides the therapeutic drug monitoring, the application of pharmacometabolomics for efficacy biomarkers potentially optimized individual patient’s treatment effects, and safety biomarkers monitoring could reduce the adverse drug reactions in real-world clinical settings.

### 4.5. Limitation and Future Analysis

The study associates pharmacokinetic profiles with the provisional dysregulated human metabolic changes that are linked to pharmacodynamic effects. Metformin did not exert glucose-lowering effects in healthy volunteers at the therapeutic dose, but one side effect was observed in this study. More cases are needed to validate the provisional safety biomarkers for adverse drug reactions. The application of a dose-dependent arm in type 2 diabetes patients and a targeted metabolomic analysis at a later phase of the clinical trial potentially narrow down the huge metabolic pathways with intense metabolite hits for a higher dose in the future investigation of metformin’s pharmacological effects.

## 5. Conclusions

In conclusion, the pharmacokinetic parameters for the single-dose oral administration of 1000-mg metformin in healthy volunteers were Cmax (1248 ng/mL) and AUC_0-infinity_ (9510 ng·h/mL) and Tmax (2.5 h); the individualized pharmacokinetics guided untargeted pharmacometabolomic of metformin, suggesting a series of human metabolic pathways, which include arginine and proline metabolism, BCAA metabolism glutathione metabolism and others that associate with metformin’s pharmacological effects of increasing insulin sensitivity and lipid metabolism. Pharmacometabolomics in early-phase clinical trials help to identify multifaceted biomarkers and understand the variation of the mechanism of action and adverse effects of a drug, which is a novel strategy for revolutionizing conventional clinical drug development toward a precision medicine approach in the future.

## Figures and Tables

**Figure 1 pharmaceutics-14-01268-f001:**
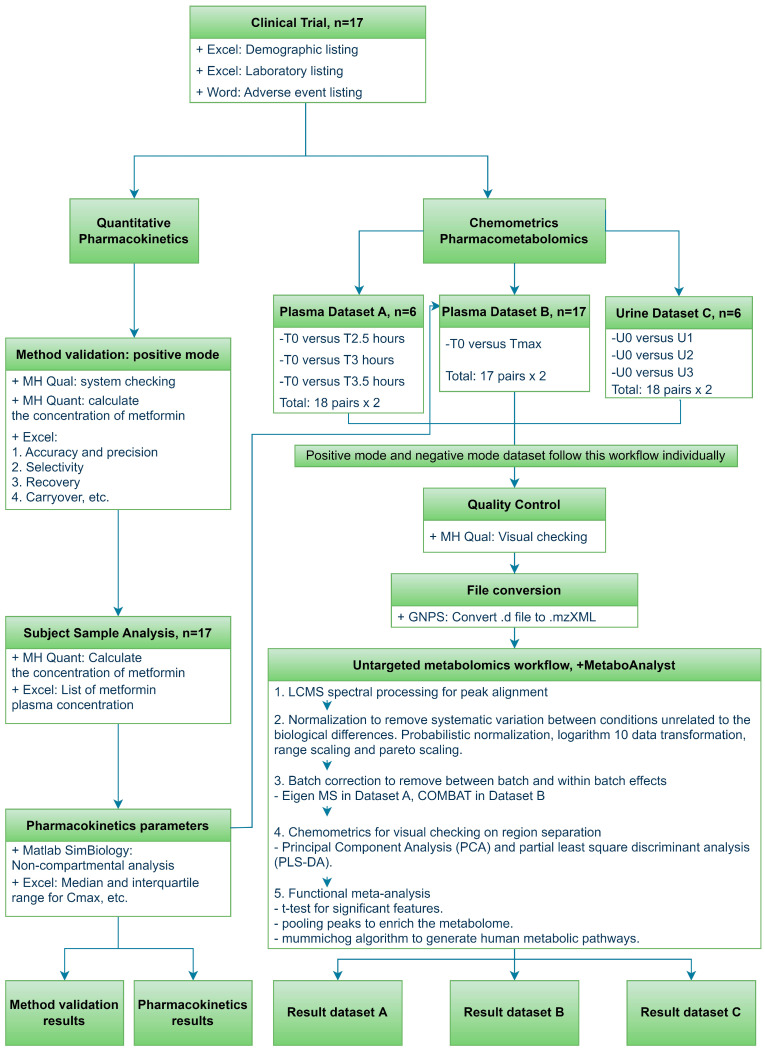
Software and statistical analysis workflow for clinical part, pharmacokinetics and pharmacometabolomics. + name the software used.

**Figure 2 pharmaceutics-14-01268-f002:**
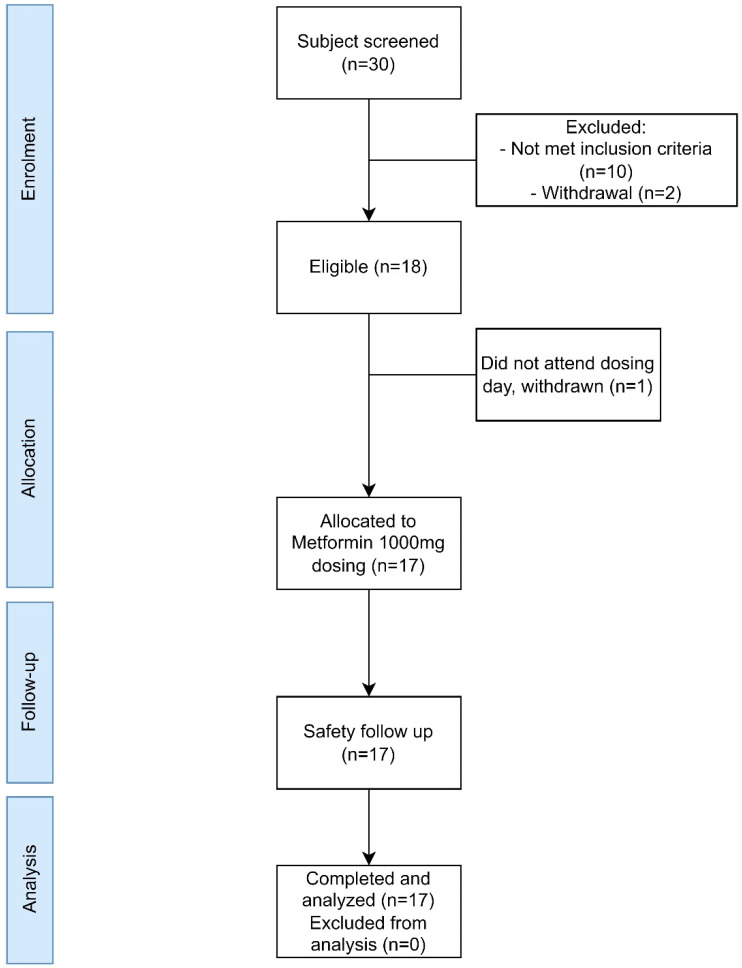
Subject disposition flow chart.

**Figure 3 pharmaceutics-14-01268-f003:**
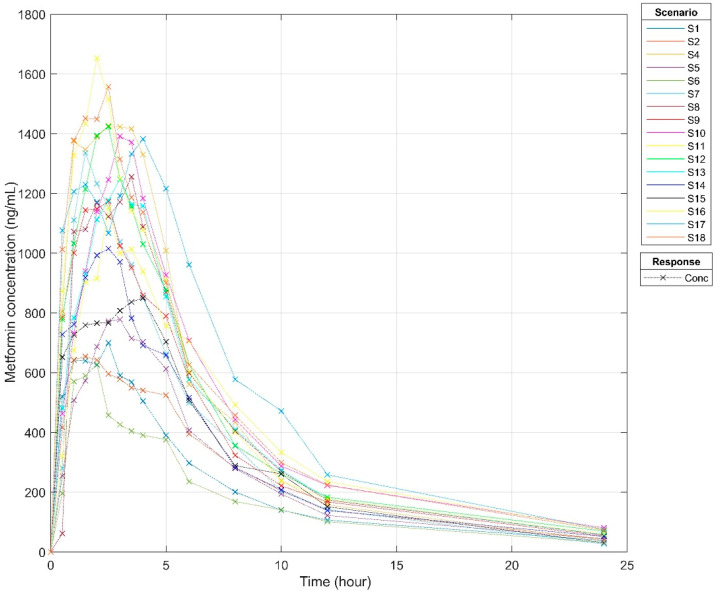
Metformin pharmacokinetic plasma concentration time curve in 17 healthy subjects. S1–S18 are 17 healthy volunteers, S3 was not present on dosing day.

**Figure 4 pharmaceutics-14-01268-f004:**
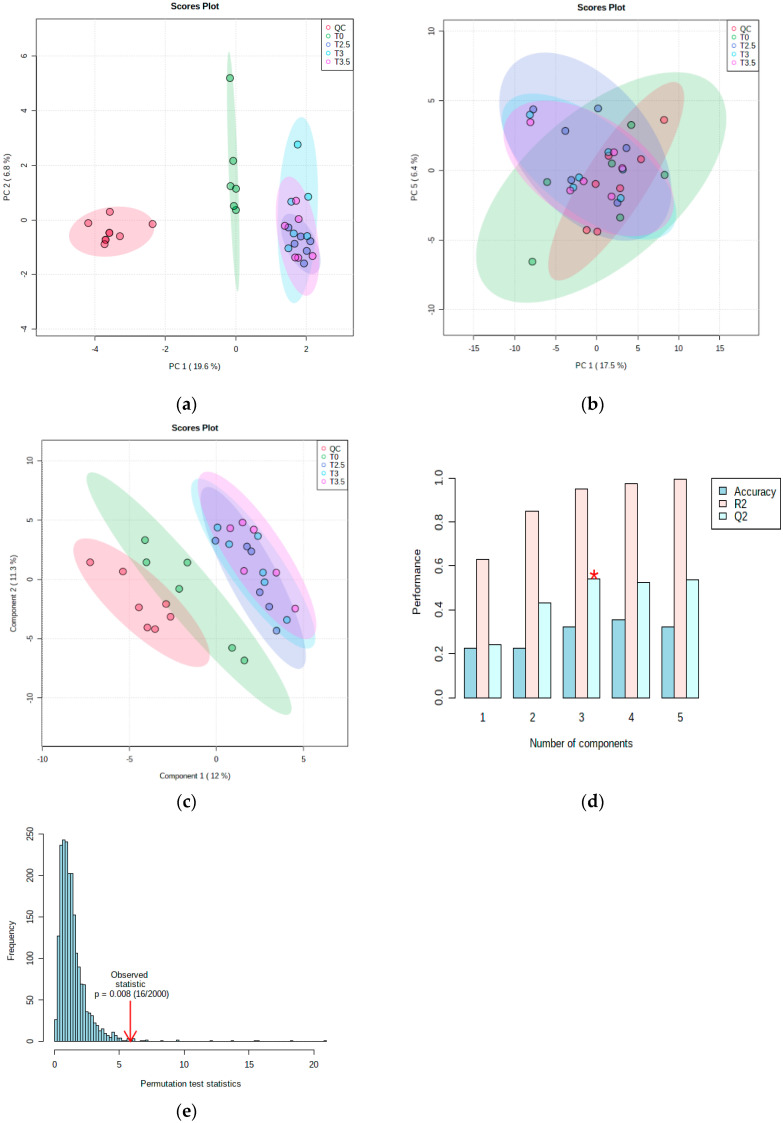
Multivariate analysis for metformin plasma dataset A: (**a**) principal component analysis (PCA) in positive mode; (**b**) PCA in negative mode; (**c**) partial least square discriminant analysis in negative mode; (**d**) cross-validation in negative mode (Q2 = 0.540, R2 = 0.948); (**e**) permutation test in negative mode (*p* < 0.05).

**Figure 5 pharmaceutics-14-01268-f005:**
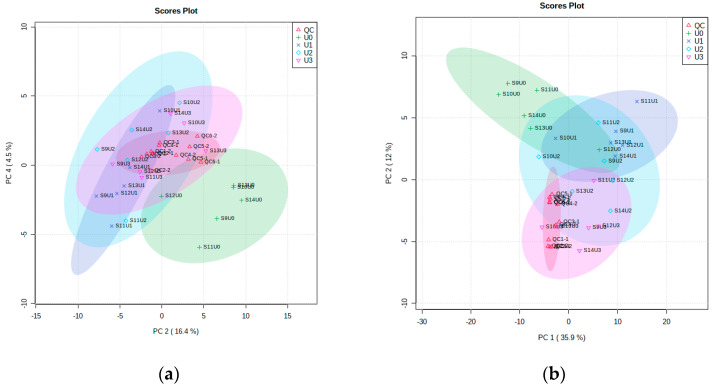
Principal component analyses (PCA) for time-points difference for features for metformin 1000-mg urine in positive mode (**a**) and negative mode in (**b**), *n* = 6.

**Figure 6 pharmaceutics-14-01268-f006:**
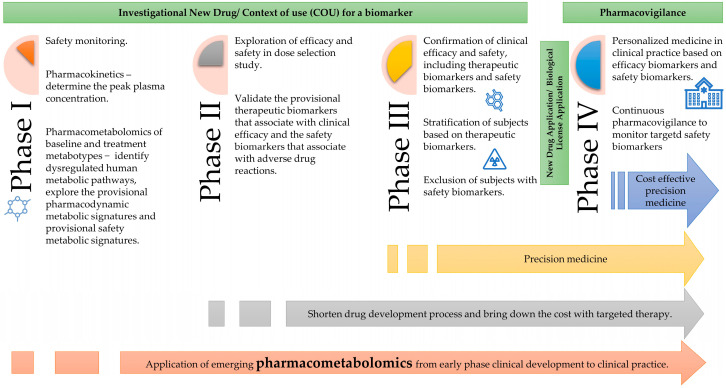
Model for application of pharmacometabolomics in clinical drug development.

**Table 1 pharmaceutics-14-01268-t001:** Demographic and clinical characteristics of the study population.

Clinical and Demographics, (*n* = 17)	Screening	Follow-Up
Ethnic, *n* (%)		
Malay	9 (52.9)	
Chinese	5 (29.4)	
Indian	2 (11.8)	
Bidayuh	1 (5.9)	
Sex, *n* (%)		
Male	17 (100.0)	
Age, mean (range), years *	25 (22–27)	
Weight, mean (range), kg *	63.9 (57.0–74.4)	
Height, mean (range), cm *	166 (165–170)	
BMI, mean (range), kg/m^2^ *	23.5 (22.1–25.0)	
Virology test		
Hepatitis Bs Ag (HbsAg)	Not detected	
Hepatitis C antibody (Anti0 HBs)	Not detected	
HIV Ag/Ab Combo	Not detected	
Biochemistry		
Sodium (mmol/L) *	140.0 (138.0–142.0)	139.0 (138.0–139.0)
Potassium (mmol/L) *	4.4 (4.2–4.6)	4.0 (3.9–4.1)
Chloride (mmol/L) *	103.0 (102.0–105.0)	103.0 (103.0–104.0)
Total CO_2_ (mmol/L) *	30.0 (30.0–31.0)	29.0 (28.0–30.0)
Anion Gap (mmol/L) *	11.0 (10.0–12.0)	10.0 (9.0–11.0)
Urea (mmol/L) *	4.9 (3.6–5.4)	4.1 (4.0–4.8)
Creatinine (µmol/L) *	82.0 (75.0–85.0)	87.0 (83.0–93.0)
Liver function test		
Albumin (g/L) *	44.0 (42.0–44.0)	39.0 (38.0–40.0)
Total bilirubin (µmol/L) *	18.0 (14.0–20.0)	11.0 (7.0–14.0)
Alkaline phosphatase (u/L) *	72.0 (63.0–80.0)	71.0 (65.0–86.0)
Alanine aminotransferase (u/l) *	21.0 (17.0–26.0)	19.0 (16.0–32.0)
Gamma GT (u/L) *	19.0 (12.0–25.0)	17.0 (12.0–22.0)
Complete blood count *		
Hemoglobin (g/L) *	160.0 (156.0–166.0)	143.0 (140.0–148.0)
Hematocrit (l/L) *	0.49 (0.47–0.49)	0.43 (0.42–0.44)
Red blood cell (10^12^/L) *	5.5 (5.4–5.9)	5.1 (5.0–5.2)
Mean corpuscular volume (fl) *	85.0 (82.0–88.0)	85.0 (84.0–87.0)
Mean corpuscular hemoglobin (pg) *	28.6 (27.2–29.7)	28.7 (27.7–29.1)
Mean corpuscular hemoglobin concentration (g/L) *	333.0 (327.0–342.0)	335.0 (329.0–346.0)
Red cell distribution width (%) *	12.2 (12.1–13.4)	12.3 (12.2–12.5)
White blood cell (10^9^/L) *	6.8 (5.7–7.1)	6.8 (6.2–8.2)
Platelet (10^9^/L) *	275.0 (247.0–319.0)	273.0 (237.0–297.0)

* Median (interquartile range).

**Table 2 pharmaceutics-14-01268-t002:** Bioanalytical validation parameters for metformin.

Parameter	Results			
Between run accuracy	LLOQ 106.71%, LQC 96.05%, MQC 99.95%, HQC 93.97%			
Between run precision	LQC 3.88, MQC 5.56, HQC 7.67			
Within run accuracy	LLOQ	LQC	MQC	HQC
Batch 1	111.17%	101.40%	102.67%	90.68%
Batch 2	99.82%	92.45%	96.23%	89.61%
Batch 3	109.15%	94.31%	100.95%	101.63%
Within run precision	LLOQ	LQC	MQC	HQC
Batch 1	0.74	0.68	3.31	2.89
Batch 2	1.82	2.87	1.62	1.94
Batch 3	5.47	3.42	2.85	3.47
Selectivity	No peak was observed at the metformin retention time for six biological batches.			
Recovery	88.58%, %CV9.85			
Carryover	No carry over is observed after 10 alternating injections of blank plasma and HQC.			
Stability	LQC CV		HQC CV	
Bench top room temperature (6 h)	−0.11		−0.09	
Three freeze-thaw cycles	−0.11		−0.23	
Auto-sampler	−0.15		−0.12	
Long-term (3 months)	0.00		−0.18	

LLOQ, lower limit of quantification; LQC, low quality control; MQC, middle quality control; HQC, high quality control; %CV, coefficient variation.

**Table 3 pharmaceutics-14-01268-t003:** Non-compartmental pharmacokinetic parameters after single-dose administration of metformin 1000 mg (*n* = 17).

Parameter	Median (Interquartile Range)
C_max (ng/mL)	1248 (849–1391)
T_max (h)	2.5 (2.5–3.0)
AUC_0_infinity_ (ng*h/mL)	9510 (7313–10,411)
AUC_0–24 (ng*h/mL)	8955 (7099–10,020)
T_half (h)	6.8 (5.5–7.0)
CL (mL/min) *	1884 (32.3)

* Mean and percentage coefficient variation, Cmax, peak plasma concentration of metformin, Tmax, time to reach Cmax; AUC_0_infinity_, total area under the plasma concentration-time curve from time zero to infinity; CL, clearance.

**Table 4 pharmaceutics-14-01268-t004:** The human metabolic pathways, total metabolite hits, significant metabolite hits for three datasets in positive mode and negative mode using pooling peaks with mummichog algorithm.

Human Metabolic Pathways (Pathway Total Metabolites in KEGG)	Dataset A, *n* = 6	Dataset B, *n* = 17	U0–U1, *n* = 6	Compound with Significant Hits (*p*-Value ≤ 0.05)
	Total Hit (Significant Hit Number, *p*-Value ≤ 0.05)			
Arginine and proline metabolism (37)	24 (4)	31 (3)	28 (5)	L-Proline ^A^; D-Proline ^A^; S-Adenosylmethioninamine ^AB^; N-Acetylputrescine ^AB^; Creatine ^BU^; Gamma-Aminobutyric acid ^U^; 4-Aminobutyraldehyde^U^; L-4-Hydroxyglutamate semialdehyde ^U^; L-Glutamic acid^U^
Glycine, serine and threonine metabolism (30)	17 (2)	23 (1)	21 (2)	Betaine aldehyde ^A^; Glyceric acid ^A^; Choline ^B^; Creatine ^BU^; Dimethylglycine ^U^
Steroid hormone biosynthesis (85)	83 (12)	84 (1)	84 (3)	Cholesterol ^A^; 20a,22b-Dihydroxycholesterol ^A^; 17alpha,20alpha-Dihydroxycholesterol ^A^; Dehydroepiandrosterone ^A^; Cortisol; 17a,21-Dihydroxy-5b-pregnane-3,11,20-trione ^A^; Testosterone ^A^; Etiocholanedione ^A^; Androstanedione ^A^; 18-Hydroxycorticosterone ^A^; 11-Dehydrocorticosterone ^A^; Tetrahydrocortisol ^A^; Testosterone glucuronide ^A^; Estrone glucuronide ^A^; Estriol-16-Glucuronide ^AB^; 11b-Hydroxyprogesterone^U^; 11b-Hydroxyandrost-4-ene-3,17-dione ^U^; 2-Methoxyestrone ^U^; 2-Methoxyestradiol ^U^; 19-Hydroxyandrost-4-ene-3,17-dione ^U^; 19-Oxoandrost-4-ene-3,17-dione ^U^; 19-Oxotestosterone ^U^; Cholesterol sulfate ^U^; 16a-Hydroxyandrost-4-ene-3,17-dione ^U^; Adrenosterone ^U^
Glutathione metabolism (19)	11 (2)	13 (1)	10 (1)	Aminopropylcadaverine ^AB^; Trypanothione disulfide ^A^; L-Glutamic acid ^U^
Galactose metabolism (27)	24 (2)	26 (1)	25 (1)	D-Gal alpha 1->6D-Gal alpha 1->6D-Glucose ^AB^; Raffinose ^AB^; D-Galactose ^U^; Alpha-D-Glucose ^U^; D-Galactose ^U^; D-Glucose ^U^; D Fructose ^U^; D-Mannose ^U^; myo-Inositol ^U^
Starch and sucrose metabolism (13)	13 (1)	13 (1)	12(1)	Dextrin ^AB^; D-Fructose ^U^; D-Glucose ^U^
Metabolism of xenobiotics by cytochrome P450 (68)	40 (5)	54 (1)	49 (3)	Glutathione episulfonium ion ^ABU^; 2-(S-Glutathionyl)acetyl chloride ^A^; Trichloroethanol glucuronide ^A^; S-(2-Chloroacetyl)glutathione ^A^; (1R)-Hydroxy-(2R)-glutathionyl-1,2-dihydronaphthalene ^A^; alpha-[3-[(Hydroxymethyl)nitrosoamino]propyl]-3-pyridinemethanol ^U^; 1-(Methylnitrosoamino)-4-(3-pyridinyl)-1,4-butanediol ^U^
Ubiquinone and other terpenoid-quinone biosynthesis (9)	9 (4)	9 (1)	9 (1)	Vitamin K1 ^AB^; Vitamin K2 ^A^; Menaquinol ^A^; Vitamin K1 2,3-epoxide ^A^; 2,3-Epoxymenaquinone ^U^
Cysteine and methionine metabolism (33)	22 (1)	28 (1)	25(1)	S-Adenosylmethioninamine ^AB^; L-Alpha-aminobutyric acid ^U^
Tryptophan metabolism (41)	23 (1)	33 (1)	36(1)	L-Tryptophan ^B^; 5-Hydroxy-N-formylkynurenine ^A^; 5-Hydroxy-L-tryptophan^U^
Aminoacyl-tRNA biosynthesis (22)	14 (1)	19 (1)	16 (2)	L-Proline ^A^; L-Tryptophan ^B^; L-Isoleucine ^U^; L-Leucine ^U^; L-Glutamic acid ^U^
Riboflavin metabolism (4)	2 (1)	3 (1)	-	Riboflavin ^AB^
Retinol metabolism (16)	16 (1)	16 (1)	-	B-Carotene ^B^; Retinoyl b-glucuronide ^AB^
Glycerophospholipid metabolism (13)	7 (1)	12 (1)	-	Acetylcholine ^A^; Choline ^B^

Note: dataset A = plasma samples for 18 pairs pre-dose versus T2.5, 3, 3.5 h data; dataset B = plasma samples from 17 pairs pre-dose versus peak-dose data; dataset U1 = urine samples 6 pairs pre-dose versus 0–4 h data; KEGG = Kyoto Encyclopedia of Gene and Genome. ^A^ = metabolites identified from dataset A, ^B^ = metabolites identified from dataset B, ^U^ = metabolites identified from dataset U1.

**Table 5 pharmaceutics-14-01268-t005:** Predicted human metabolic pathways based on mummichog algorithm for metformin 1000 mg at time-point U0 versus U1, U0 versus U2 and U0 versus U3 with the number of pathway metabolites, total metabolite hits and significant metabolite hits (*p*-value ≤ 0.005).

Human Metabolic Pathways	Pathway Total Metabolites/Total Metabolites Hit (Significant Metabolites Hit, *p* ≤ 0.005)			Compound with Significant Hits (*p*-Value ≤ 0.05)
	U0 vs. U1	U0 vs. U2	U0 vs. U3	
Arginine and proline metabolism (37)	28 (5)	26 (2)	-	Creatine ^U1U2^; Gamma-Aminobutyric acid ^U1^; 4-Aminobutyraldehyde ^U1^; L-4-Hydroxyglutamate semialdehyde ^U1U2^; L-Glutamic acid ^U1U2^
Glycine, serine and threonine metabolism (30)	21 (2)	22 (1)	22 (1)	Creatine ^U1U2^; Dimethylglycine ^U1^; L-2-Amino-3-oxobutanoic acid ^U3^
Glycosaminoglycan degradation (21)	9 (2)	-	-	(GalNAc)2 (GlcA)1 (S)1 ^U1^; (GlcA)2 (GlcNAc)1 (S)2 ^U1^; DWA-2 ^U1^
Drug metabolism—cytochrome P450 (43)	38 (4)	-	-	Alcophosphamide ^U1^; Codeine-6-glucuronide ^U1^; Citalopram N-oxide ^U1^; L-alpha-Acetyl-N,N-dinormethadol ^U1^
Butanoate metabolism (15)	9 (2)	9 (1)	-	2-Hydroxyglutarate ^U1^; Gamma-Aminobutyric acid ^U1^; L-Glutamic acid ^U1U2^
Arginine biosynthesis (14)	10 (1)	9 (1)	-	L-Glutamic acid ^U1U2^

Note: dataset U0 vs. U1 = urine samples 6 pairs data (pre-dose versus 0–4 h); dataset U0 vs. U2 = urine samples 6 pairs data (pre-dose versus 4–8 h); dataset U0 vs. U3 = urine samples 6 pairs data (pre-dose versus 8–12 h), ^U1^ = metabolites found in dataset U0vsU1, ^U2^ = metabolites found in dataset U0 vs. U2, ^U3^ = metabolites found in dataset U0 vs. U3.

## Data Availability

Not applicable.

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
