# Peer review of "Pharmacokinetic–Pharmacometabolomic Approach in Early-Phase Clinical Trials: A Way Forward for Targeted Therapy in Type 2 Diabetes"

_pharmaceutics, 2022, doi:10.3390/pharmaceutics14061268_

Round 1

Reviewer 1 Report

The manuscript could be interesting, sometimes it seems to be too long and dispersive, such as in the results and discussion section. On the contrary, introduction could be modified including more references and more detailed  of the study. A scheme of the applied stragety could be added. 

Figure 1 is difficult to read.

Too many tables along the manuscript. Some of this information could be moved to the supporting information.

Captions to figures should be placed after the related images.

Reviewer 2 Report

A very intesting study, that conveys us lots of infos. From a biostatistics and PK/PD point of view, I have some suggestions for the Authors:

- line 35, please add single-dose just before 1000 mg

- line 43 and everywhere, please report any descriptive stats for continuous variables always as median(IQR) and not mean/SD

- line 43 and everywhere, please report AUC as ng*h/ml

- line 97, please define AMPK

- line 119, please specify fasting conditions (timing...)

- line 216, median Tmax (see my comment for line 43), here and everywhere

- line 235,  the use of PCA should be better defined (e.g. exploratory data analysis, making decisions in predictive models, dimensionality reduction and so on), why and how have you applied this method!? for what!? how have you selected al, the covariated coming from univariate models to run them together in the multivariate one!?

- line 254, Normalization... I'm not sure that to normalize such a small ans asymmetric sample could ensure a good reliability, what's your opinion!?

- line 292, a formal and classical stats method section is totally lacking (e.g. descriptive stats, inferential stats, uni- and multi- variate modeling techniques, p-value estimation techniques, softwares...). It seems that the Authors have applied a totally automatic instrumental routine, more than a full stats plan

-table 1, what about the potential PK/PD confounding role of the different 4 ethnicities? Anyway, race is an outdated terminology

- table 3, rather than CV, you should report median(IQR) values for any parameter

- line 362, please review my comment at line 235, the reader must be informed about the meaning of this specific PCA usage

-figure 4, please read and apply the previous comment

- line 468, it's important to harmonize Cmax unit measures like ng/ml and micromoles, here and all around the manuscript

Reviewer 3 Report

The study is straightforward, methods are written in great detail, data are convincing, article reads well throughout the manuscript. There were very few grammatical mistakes, and sentence formation was good and easy to understand. The sample size was low in the study (n=17). Authors made a good attempt to write abbreviations. There are 2 mistakes which can be easily corrected-

Line 508- period after [35]

Line 525- incomplete sentence.

The study by Khim Boon Tee et al is clinically significant with pharmacokinetic parameters for single-dose oral administration of  oral 1000mg metformin in healthy volunteers were Cmax (1154 ng/ml) and AUC0-infinity (8846 644 h.ng/ml), Tmax (2.65 hours), individualized pharmacokinetics guided untargeted pharmacometabolomic of metformin suggests a series of human metabolic pathways which include arginine and proline metabolism, BCAA metabolism glutathione metabolism and  others that associates with metformin pharmacological effects of increasing insulin sensitivity and lipid metabolism. Pharmacometabolomic has the potential to identify multifaceted biomarkers and understand the variation of the mechanism of action and side effects of a drug which can be individualized towards each patient for a better outcome. The materials and methods section of the manuscript is written in detail, which improved the quality of the manuscript.

Round 2

Reviewer 2 Report

The Authors were able to address the vast majority of the current concerns. Figure 1 is a mere uncommented list, while a full and detailed stats chapter/methodological section would have been very useful (e.g., we can not perform any multivariate model before having estimated all the univariate ones! no results of them have been reported). In table 1, please update median(IQR) values. 
